# Active Retrieval Augmented Generation

**Zhengbao Jiang**[1*]   **Frank F. Xu**[1*]   **Luyu Gao**[1*]   **Zhiqing Sun**[1*]   **Qian Liu**[2]
**Jane Dwivedi-Yu**[3]   **Yiming Yang**[1]   **Jamie Callan**[1]   **Graham Neubig**[1]
[1]Language Technologies Institute, Carnegie Mellon University
[2]Sea AI Lab   [3]FAIR, Meta
{zhengbaj,fangzhex,luyug,zhiqings,gneubig}@cs.cmu.edu

## Abstract

Despite the remarkable ability of large language models (LMs) to comprehend and generate language, they have a tendency to hallucinate and create factually inaccurate output. Augmenting LMs by retrieving information from external knowledge resources is one promising solution. Most existing retrieval augmented LMs employ a retrieve-and-generate setup that only retrieves information once based on the input. This is limiting, however, in more general scenarios involving generation of long texts, where continually gathering information throughout generation is essential. In this work, we provide a generalized view of *active retrieval augmented generation*, methods that actively decide when and what to retrieve across the course of the generation. We propose **F**orward-**L**ooking **A**ctive **RE**trieval augmented generation (**FLARE**), a generic method which iteratively uses a prediction of the upcoming sentence to anticipate future content, which is then utilized as a query to retrieve relevant documents to regenerate the sentence if it contains low-confidence tokens. We test FLARE along with baselines comprehensively over 4 long-form knowledge-intensive generation tasks/datasets. FLARE achieves superior or competitive performance on all tasks, demonstrating the effectiveness of our method.[1]

## 1 Introduction

Generative language models (LMs) (Brown et al., 2020; Ouyang et al., 2022; OpenAI, 2023; Chowdhery et al., 2022; Zhang et al., 2022; Touvron et al., 2023; Zhao et al., 2023) have become a foundational component in natural language processing (NLP) systems with their remarkable abilities. Although LMs have memorized some world knowledge during training (Petroni et al., 2019; Roberts et al., 2020; Jiang et al., 2020), they still tend to

hallucinate and create imaginary content (Maynez et al., 2020; Zhou et al., 2021). Augmenting LMs with retrieval components that look up relevant information from external knowledge resources is a promising direction to address hallucination (Khandelwal et al., 2020; Izacard et al., 2022).

Retrieval augmented LMs commonly use a retrieve-and-generate setup where they retrieve documents based on the user's input, and then generate a complete answer conditioning on the retrieved documents (Chen et al., 2017; Guu et al., 2020; Lewis et al., 2020; Izacard and Grave, 2021; Sachan et al., 2021; Lee et al., 2021; Jiang et al., 2022; Izacard et al., 2022; Nakano et al., 2021; Qian et al., 2023; Lazaridou et al., 2022; Shi et al., 2023). These single-time retrieval augmented LMs outperform purely parametric LMs, particularly for short-form knowledge-intensive generation tasks such as factoid question answering (QA) (Kwiatkowski et al., 2019; Joshi et al., 2017), where *the information needs are clear in the user's input, and it is sufficient to retrieve relevant knowledge once solely based on the input*.

Increasingly powerful large LMs have also demonstrated abilities in more complex tasks that involve generating long-form output, such as long-form QA (Fan et al., 2019; Stelmakh et al., 2022), open-domain summarization (Cohen et al., 2021; Hayashi et al., 2021; Giorgi et al., 2022), and (chain-of-thought; CoT) reasoning (Wei et al., 2022; Ho et al., 2020; Geva et al., 2021; Hendrycks et al., 2020). In contrast to short-form generation, long-form generation presents complex information needs that are *not always evident from the input alone*. Similar to how humans gradually gather information as we create content such as papers, essays, or books, long-form generation with LMs would *require gathering multiple pieces of knowledge throughout the generation process*. For example, to generate a summary about a particular topic, the initial retrieval based on the topic name

---

[*]Lead contributors.

[1]Code and datasets are available at https://github.com/jzbjyb/FLARE.

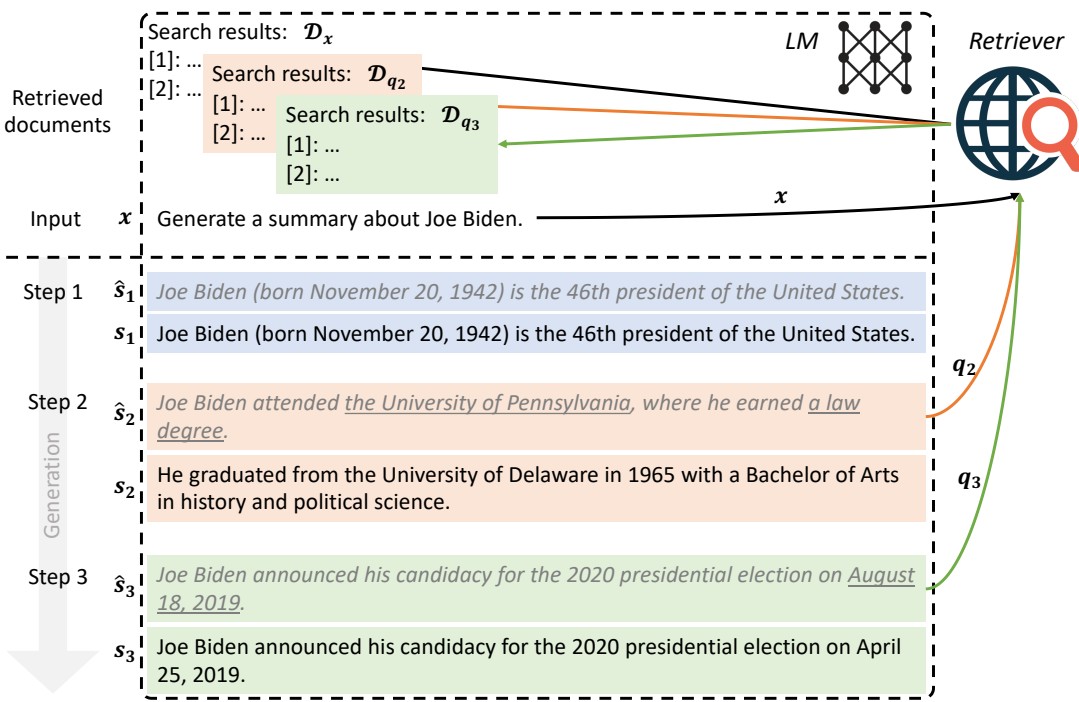

Figure 1: An illustration of forward-looking active retrieval augmented generation (FLARE). Starting with the user input $x$ and initial retrieval results $\mathcal{D}_x$, FLARE iteratively generates a temporary next sentence (shown in *gray italic*) and check whether it contains low-probability tokens (indicated with underline). If so (step 2 and 3), the system retrieves relevant documents and regenerates the sentence.

(e.g., Joe Biden) may not cover all aspects and details. It is crucial to retrieve extra information as needed during generation, such as when generating a certain aspect (e.g., Joe Biden's education history) or a specific detail (e.g., the date of Joe Biden's presidential campaign announcement).

Several attempts have been made to retrieve multiple times throughout generation. These attempts include methods that passively use the past context to retrieve additional information at a fixed interval (Khandelwal et al., 2020; Borgeaud et al., 2022; Ram et al., 2023; Trivedi et al., 2022) which might not accurately reflect what LMs intend to generate in the future or retrieve at inappropriate points. Some works in multihop QA decompose the full question into sub-questions, each of which is used to retrieve extra information (Press et al., 2022; Yao et al., 2022; Khot et al., 2022; Khattab et al., 2022).

We ask the following question: can we create a simple and generic retrieval augmented LM that *actively decides when and what to retrieve* throughout the generation process, and are applicable to a variety of long-form generation tasks? We provide a generalized view of active retrieval augmented generation. Our hypothesis regarding *when to retrieve* is that LMs should retrieve information only when

they lack the required knowledge to avoid unnecessary or inappropriate retrieval that occurs in passive retrieval augmented LMs (Khandelwal et al., 2020; Borgeaud et al., 2022; Ram et al., 2023; Trivedi et al., 2022). Given the observation that large LMs tend to be well-calibrated and low probability/confidence often indicates a lack of knowledge (Kadavath et al., 2022), we adopt an active retrieval strategy that only retrieves when LMs generate low-probability tokens. When deciding *what to retrieve*, it is important to consider what LMs intend to generate in the future, as the goal of active retrieval is to benefit future generations. Therefore, we propose anticipating the future by generating a temporary next sentence, using it as a query to retrieve relevant documents, and then regenerating the next sentence conditioning on the retrieved documents. Combining the two aspects, we propose **F**orward-**L**ooking **A**ctive **RE**trieval augmented generation (**FLARE**), as illustrated in Figure 1. FLARE iteratively generates *a temporary next sentence*, use it as the query to retrieve relevant documents *if it contains low-probability tokens* and regenerate the next sentence until reaches the end.

FLARE is applicable to any existing LMs at inference time without additional training. Con-

sidering the impressive performance achieved by GPT-3.5 (Ouyang et al., 2022) on a variety of tasks, we examine the effectiveness of our methods on text-davinci-003. We evaluate FLARE on 4 diverse tasks/datasets involving generating long outputs, including multihop QA (2WikiMultihopQA), commonsense reasoning (StrategyQA), long-form QA (ASQA), and open-domain summarization (WikiAsp) (Ho et al., 2020; Geva et al., 2021; Stelmakh et al., 2022; Hayashi et al., 2021). Over all tasks, FLARE achieves superior or competitive performance compared to single-time and multi-time retrieval baselines, demonstrating the effectiveness and generalizability of our method.

## 2 Retrieval Augmented Generation

We formally define single-time retrieval augmented generation and propose the framework of active retrieval augmented generation.

### 2.1 Notations and Definitions

Given a user input $x$ and a document corpus $\mathcal{D} = \{d_i\}_{i=1}^{|\mathcal{D}|}$ (such as all Wikipedia articles), the goal of retrieval augmented LMs is to generate the answer $y = [s_1, s_2, ..., s_m] = [w_1, w_2, ..., w_n]$ containing $m$ sentences or $n$ tokens leveraging information retrieved from the corpus.

In retrieval augmented LM, the LM typically pairs with a retriever that can retrieve a list of documents $\mathcal{D}_q = \text{ret}(q)$ for a query $q$; the LM conditions on both the user input $x$ and retrieved documents $\mathcal{D}_q$ to generate the answer. Since we focus on examining various methods of determining when and what to retrieve, we follow existing methods (Ram et al., 2023; Trivedi et al., 2022) to prepend the retrieved documents before the user input to aid future generation for both baselines and our method for fair comparisons: $y = \text{LM}([\mathcal{D}_q, x])$, where $[\cdot, \cdot]$ is concatenation following the specified order.

### 2.2 Single-time Retrieval Augmented Generation

The most common choice is to directly use the user input as the query for retrieval and generate the complete answer at once $y = \text{LM}([\mathcal{D}_x, x])$.

### 2.3 Active Retrieval Augmented Generation

To aid long-form generation with retrieval, we propose active retrieval augmented generation. It is a generic framework that actively decides when and what to retrieve through the generation process,

resulting in the interleaving of retrieval and generation. Formally, at step $t (t \geq 1)$, the retrieval query $q_t$ is formulated based on both the user input $x$ and previously generated output $y_{<t} = [y_0, ..., y_{t-1}]$:

$$q_t = \text{qry}(x, y_{<t}),$$

where $\text{qry}(\cdot)$ is the query formulation function. At the beginning ($t = 1$), the previous generation is empty ($y_{<1} = \emptyset$), and the user input is used as the initial query ($q_1 = x$). Given retrieved documents $\mathcal{D}_{q_t}$, LMs continually generate the answer until the next retrieval is triggered or reaches the end:

$$y_t = \text{LM}([\mathcal{D}_{q_t}, x, y_{<t}]),$$

where $y_t$ represents the generated tokens at the current step $t$, and the input to LMs is the concatenation of the retrieved documents $\mathcal{D}_{q_t}$, the user input $x$, and the previous generation $y_{<t}$. We discard previously retrieved documents $\cup_{t'<t}\mathcal{D}_{q_{t'}}$ and only use the retrieved documents from the current step to condition the next generation to prevent reaching the input length limit of LMs.

## 3 FLARE: Forward-Looking Active REtrieval Augmented Generation

Our intuition is that (1) LMs should only retrieve information when they do not have the necessary knowledge to avoid unnecessary or inappropriate retrieval, and (2) the retrieval queries should reflect the intents of future generations. We propose two forward-looking active retrieval augmented generation (FLARE) methods to implement the active retrieval augmented generation framework. The first method prompts the LM to generate retrieval queries when necessary while generating the answer using retrieval-encouraging instructions, denoted as FLARE_instruct. The second method directly uses the LM's generation as search queries, denoted as FLARE_direct, which iteratively generates the next sentence to gain insight into the future topic, and if uncertain tokens are present, retrieves relevant documents to regenerate the next sentence.

### 3.1 FLARE with Retrieval Instructions

Inspired by Toolformer (Schick et al., 2023), a straightforward way of expressing information needs for retrieval is to generate "[Search(query)]" when additional information is needed (Schick et al., 2023), e.g., "The colors on the flag of Ghana have the following meanings. Red is for [Search(Ghana flag red meaning)] the blood of martyrs, ..." When working with GPT-3.5 models that

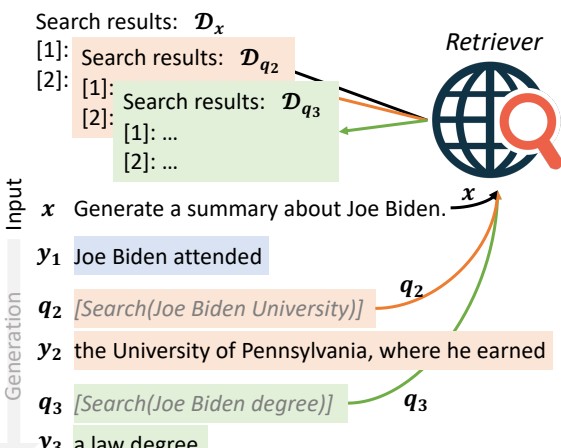

Figure 2: An illustration of forward-looking active retrieval augmented generation with retrieval instructions (FLARE_instruct). It iteratively generates search queries (shown in *gray italic*) to retrieve relevant information to aid future generations.

offer only API access, we elicit such behavior by few-shot prompting (Brown et al., 2020).

Specifically, for a downstream task, we place the search-related instruction and exemplars at the beginning as skill 1, followed by the instruction and exemplars of the downstream task as skill 2. Given a test case, we ask LMs to combine skills 1 and 2 to generate search queries while performing the task. The structure of the prompt is shown in Prompt 3.1, and full details can be found in Prompt D.3.

---

**Prompt 3.1: retrieval instructions**

Skill 1. An instruction to guide LMs to generate search queries.
Several search-related exemplars.

Skill 2. An instruction to guide LMs to perform a specific downstream task (e.g., multihop QA).
Several task-related exemplars.

An instruction to guide LMs to combine skills 1 and 2 for the test case.
The input of the test case.

---

As shown in Figure 2, when the LM generates "[Search(query)]" (shown in *gray italic*), we stop the generation and use the query terms to retrieve relevant documents, which are prepended before the user input to aid future generation until the next search query is generated or reaches the end. Additional implementation details are included in Appendix A.

## 3.2 Direct FLARE

Since we cannot fine-tune black-box LMs, we found queries generated by FLARE_instruct through

retrieval instructions might not be reliable. Therefore, we propose a more direct way of forward-looking active retrieval that uses the next sentence to decide when and what to retrieve.

### 3.2.1 Confidence-based Active Retrieval

As shown in Figure 1, at step $t$, we first generate a temporary next sentence $\hat{s}_t = \text{LM}([\boldsymbol{x}, \boldsymbol{y}_{<t}])$ without conditioning on retrieved documents. Then we decide whether to trigger retrieval and formulate queries based on $\hat{s}_t$. If the LM is confident about $\hat{s}_t$, we accept it without retrieving additional information; if not, we use $\hat{s}_t$ to formulate search queries $\boldsymbol{q}_t$ to retrieve relevant documents, and then regenerate the next sentence $\boldsymbol{s}_t$. The reason we utilize sentences as the basis of our iteration is due to their significance as semantic units that are neither too short nor too lengthy like phrases and paragraphs. However, our approach can also utilize phrases or paragraphs as the basis.

Since LMs tend to be well-calibrated that low probability/confidence often indicates a lack of knowledge (Jiang et al., 2021; Kadavath et al., 2022; Varshney et al., 2022), we actively trigger retrieval if any token of $\hat{s}_t$ has a probability lower than a threshold $\theta \in [0, 1]$. $\theta = 0$ means retrieval is never triggered, while $\theta = 1$ triggers retrieval every sentence.

$$\boldsymbol{y}_t = \begin{cases} \hat{s}_t & \text{if all tokens of } \hat{s}_t \text{ have probs} \geq \theta \\ \boldsymbol{s}_t = \text{LM}([\mathcal{D}_{\boldsymbol{q}_t}, \boldsymbol{x}, \boldsymbol{y}_{<t}]) & \text{otherwise} \end{cases}$$

where the query $\boldsymbol{q}_t$ is formulated based on $\hat{s}_t$.

### 3.2.2 Confidence-based Query Formulation

One way to perform retrieval is to directly use the next sentence $\hat{s}_t$ as the query $\boldsymbol{q}_t$. This shares a similar spirit with methods that use generated hypothetical titles or paragraphs from LMs as retrieval queries or evidences (Gao et al., 2022; Sun et al., 2022; Yu et al., 2022; Mao et al., 2021). We generalize such techniques to long-form generation where active information access is essential.

We found retrieving with the next sentence achieves significantly better results than with the previous context, as shown later in subsection 6.2. However, it has a risk of perpetuating errors contained in it. For example, if the LM produces the sentence "Joe Biden attended the University of Pennsylvania" instead of the correct fact that he attended the University of Delaware, using this erroneous sentence as a query might retrieve mislead-

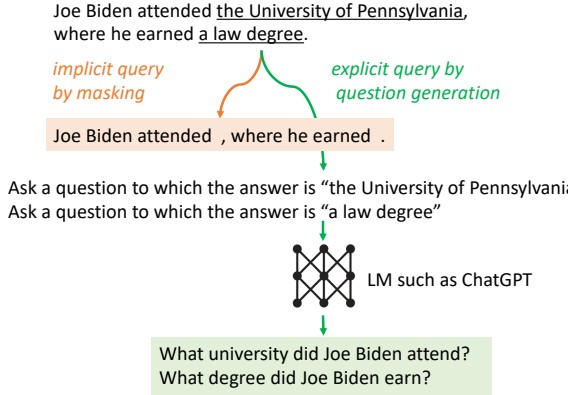

Figure 3: Implicit and explicit query formulation. Tokens with low probabilities are marked with underlines.

ing information. We propose two simple methods to overcome this issue as illustrated in Figure 3.

**Masked sentences as implicit queries.** The first method masks out low-confidence tokens in $\hat{s}_t$ with probabilities below a threshold $\beta \in [0, 1]$, where a higher $\beta$ results in more aggressive masking. This removes potential distractions from the sentence to improve retrieval accuracy.

**Generated questions as explicit queries.** Another method is to generate explicit questions that target the low-confident span in $\hat{s}_t$. For example, if the LM is uncertain about "the University of Pennsylvania", a question like "Which university did Joe Biden attend?" can help retrieve relevant information. Self-ask (Press et al., 2022) achieved this by manually inserting follow-up questions into downstream task exemplars as shown later in Prompt D.2, which requires task-specific annotation efforts. Instead, we developed a universal approach that generates questions for low-confidence spans without additional annotation. Specifically, We first extract all spans from $\hat{s}_t$ with probabilities below $\beta$. For each extracted span $z$, we prompt gpt-3.5-turbo to generate a question $q_{t,z}$ that can be answered with the span:

> **Prompt 3.2: zero-shot question generation**
>
> User input $x$.
> Generated output so far $y_{\leq t}$.
>
> Given the above passage, ask a question to which the answer is the term/entity/phrase "$z$".

We retrieve using each generated question and interleave the returned documents into a single ranking list to aid future generations. In summary,

queries $q_t$ are formulated based on $\hat{s}_t$ as follows:

$$q_t = \begin{cases} \emptyset & \text{if all tokens of } \hat{s}_t \text{ have probs} \geq \theta \\ \text{mask}(\hat{s}_t) \text{ or qgen}(\hat{s}_t) & \text{otherwise} \end{cases}$$

### 3.3 Implementation Details

**Base LM** We validate our method on one of the most advanced GPT-3.5 LMs text-davinci-003 by iteratively querying their API.[2]

**Document corpus and retrievers.** Since we focus on the integration of retrieval and generation, we use off-the-shelf retrievers that take queries as inputs and return a list of relevant documents. For datasets that mainly rely on knowledge from Wikipedia, we use the Wikipedia dump from Karpukhin et al. (2020) and employ BM25 (Robertson and Zaragoza, 2009) as the retriever. For datasets that rely on knowledge from the open web, we use the Bing search engine as our retriever.[3]

**Retrieved document formatting.** Multiple retrieved documents are linearized according to their ranking and then added to the beginning of the user input using Prompt D.1.

Other implementation details such as sentence tokenization and efficiency are included Appendix A.

## 4 Multi-time Retrieval Baselines

Existing passive multi-time retrieval augmented LMs can also be formulated using our framework (subsection 2.3). In this section, we formally introduce three baseline categories based on when and what to retrieve. These baselines are not exact reproductions of the corresponding paper because many design choices differ which makes direct comparisons impossible. We implemented them using the same settings, with the only variation being when and what to retrieve.

**Previous-window** approaches trigger retrieval every $l$ tokens, where $l$ represents the window size. Generated tokens from the previous window are used as the query:

$$q_t = y_{t-1} \quad (t \geq 2),$$
$$y_t = [w_{(t-1)l+1}, ..., w_{tl}].$$

Some existing methods in this category are RETRO (Borgeaud et al., 2022), IC-RALM (Ram et al.,

---

[2]https://api.openai.com/v1/completions April 23.
[3]https://www.microsoft.com/en-us/bing/apis/bing-web-search-api

2023), which retrieve every few tokens, and KNN-LM (Khandelwal et al., 2020), which retrieves every token.[4] We follow Ram et al. (2023) to use a window size of $l = 16$.

**Previous-sentence** approaches trigger retrieval every sentence and use the previous sentence as the query, and IRCoT (Trivedi et al., 2022) belongs to this category:

$$q_t = y_{t-1} \quad (t \geq 2),$$
$$y_t = s_t.$$

**Question decomposition** approaches manually annotated task-specific exemplars to guide LMs to generate decomposed sub-questions while producing outputs. For example, self-ask (Press et al., 2022), a method in this category, manually inserts sub-questions in exemplars using Prompt D.2. For the test case, retrieval is triggered dynamically whenever the model generates a sub-question.

The aforementioned approaches can retrieve additional information while generating. However, they have notable drawbacks: (1) Using previously generated tokens as queries might not reflect what LMs intend to generate in the future. (2) Retrieving information at a fixed interval can be inefficient because it might occur at inappropriate points. (3) Question decomposition approaches require task-specific prompt engineering, which restricts their generalizability in new tasks.

## 5 Experimental Setup

We evaluate the effectiveness of FLARE on 4 diverse knowledge-intensive tasks using few-shot in-context learning (Radford et al., 2019; Brown et al., 2020; Liu et al., 2023). We follow previous works (Trivedi et al., 2022) to sub-sample at most 500 examples from each dataset due to the cost of running experiments. Datasets, metrics, and settings are summarized in Table 7 of Appendix B. The hyperparameters of FLARE are selected based on the development set and listed in Table 9. FLARE refers to FLARE$_{\text{direct}}$ if not specifically stated.

**Multihop QA** The goal of multihop QA is to answer complex questions through information retrieval and reasoning. We use 2WikiMultihopQA (Ho et al., 2020) which contains 2-hop complex

---

[4]Since KNN-LM uses the contextualized representation corresponding to the current decoding position to retrieve relevant information which encodes all previous tokens. Strictly speaking, $q_t$ should be $y_{<t}$.

questions sourced from Wikipedia articles that require composition, comparison, or inference, e.g., "Why did the founder of Versus die?" We follow Wang et al. (2022) to generate both the chain-of-thought and the final answer. Experimental setting details are included in Appendix B.

We use regular expressions to extract the final answer from the output and compare it with the reference answer using exact match (EM), and token-level $F_1$, precision, and recall.

**Commonsense reasoning** Commonsense reasoning requires world and commonsense knowledge to generate answers. We use StrategyQA (Geva et al., 2021) which is a collection of crowdsourced yes/no questions, e.g., "Would a pear sink in water?" We follow Wei et al. (2022) to generate both the chain-of-thought and the final yes/no answer. Details are included in Appendix B.

We extract the final answer and match it against the gold answer using exact match.

**Long-form QA** Long-form QA aims to generate comprehensive answers to questions seeking complex information (Fan et al., 2019; Stelmakh et al., 2022). We use ASQA (Stelmakh et al., 2022) as our testbed where inputs are ambiguous questions with multiple interpretations, and outputs should cover all of them. For example, "Where do the Philadelphia Eagles play their home games?" could be asking about the city, sports complex, or stadium. We found in many cases it is challenging even for humans to identify which aspect of the question is ambiguous. Therefore, we created another setting (ASQA-hint) where we provide a brief hint to guide LMs to stay on track when generating answers. The hint for the above case is "This question is ambiguous in terms of which specific location or venue is being referred to." Experimental setting details are included in Appendix B.

We use metrics from Stelmakh et al. (2022), including EM, RoBERTa-based QA score (Disambig-$F_1$), ROUGE (Lin, 2004), and an overall score combining Disambig-$F_1$ and ROUGE (DR).

**Open-domain summarization** The goal of open-domain summarization is to generate a comprehensive summary about a topic by gathering information from open web (Giorgi et al., 2022). We use WikiAsp (Hayashi et al., 2021) which aims to generate aspect-based summaries about entities from 20 domains in Wikipedia, e.g., "Generate a summary about Echo School (Oregon) including the

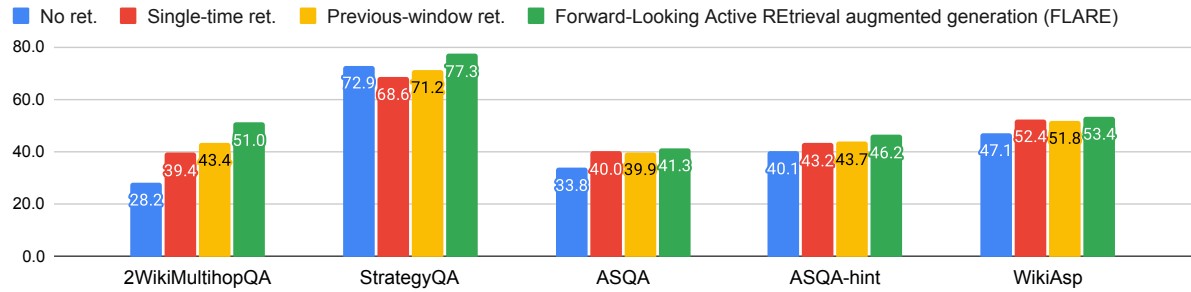

Figure 4: Comparision between FLARE and baselines across all tasks/datasets. We report the primary metric for each dataset: EM for 2WikiMultihopQA, StrategyQA, and ASQA, and UniEval for WikiAsp.

following aspects: academics, history." Experimental setting details are included in Appendix B.

Metrics include ROUGE, named entity-based $F_1$, and UniEval (Zhong et al., 2022) which measures factual consistency.

## 6 Experimental Results

We first report overall results across 4 tasks/datasets and compare the performance of FLARE with all the baselines introduced in section 4. We then run ablation experiments to study the efficacy of various design choices of our method.

### 6.1 Comparison with Baselines

**Overall results.** The overall performance of FLARE and baseline across all tasks/datasets are reported in Figure 4. FLARE outperforms all baseline on all tasks/datasets, indicating that FLARE is a generic method that can effectively retrieve additional information throughout the generation.

Among various tasks, multihop QA shows the most significant improvement. This is largely due to the task's clear definition and specific objective of producing the final answer through a 2-hop reasoning process, which makes it easier for LMs to generate on-topic output. In contrast, ASQA and WikiAsp are more open-ended, which increases the difficulty of both generation and evaluation. The improvement on ASQA-hint is larger than that of ASQA because identifying ambiguous aspects is challenging even for humans in many cases, and providing a generic hint helps LMs to stay on topic.

**Thorough comparisons with baselines.** The performance of all baselines on 2WikiMultihopQA are reported in Table 1. FLARE outperforms all baselines by a large margin, which confirms that forward-looking active retrieval is highly effective. Most multi-time retrieval augmented approaches outperform single-time retrieval but with different

| Methods | EM | $F_1$ | Prec. | Rec. |
|---|---|---|---|---|
| No retrieval | 28.2 | 36.8 | 36.5 | 38.6 |
| Single-time retrieval | 39.4 | 48.8 | 48.6 | 51.5 |
| *Multi-time retrieval* | | | | |
| Previous-window | 43.2 | 52.3 | 51.7 | 54.5 |
| Previous-sentence | 39.0 | 49.2 | 48.9 | 51.8 |
| Question decomposition | 47.8 | 56.4 | 56.1 | 58.6 |
| FLARE$_{instruct}$ (ours) | 42.4 | 49.8 | 49.1 | 52.5 |
| FLARE$_{direct}$ (ours) | **51.0** | **59.7** | **59.1** | **62.6** |

Table 1: FLARE and baselines on 2WikiMultihopQA. Previous-window (Borgeaud et al., 2022; Ram et al., 2023), previous-sentence (Trivedi et al., 2022), and question decomposition (Press et al., 2022; Yao et al., 2022) methods are reimplemented for fair comparisons.

margins. The improvement of retrieving using the previous sentence is relatively small which we hypothesize is mainly because the previous sentence often describes entities or relations different from those in the next sentence in 2WikiMultihopQA. While the previous-window approach might use the first half of a sentence to retrieve information potentially helpful for generating the second half. Among all baselines, the question decomposition approach (Press et al., 2022) achieves the best performance. which is not surprising since the in-context exemplars manually annotated with decomposed sub-questions (Prompt D.2) guide LMs to generate sub-questions that align with the topic/intent of future generations. FLARE outperforms this baseline, indicating that manual exemplar annotation is not necessary for effective future-aware retrieval. The gap between FLARE$_{instruct}$ and question decomposition is large, indicating that teaching LMs to generate search queries using task-generic retrieval instructions and exemplars is challenging.

We report all metrics for the other datasets in Table 2. FLARE outperforms baselines with respect to all metrics. Retrieval using the previ-

| Datasets Metrics | StrategyQA EM | ASQA EM D-F$_1$ R-L DR | ASQA-hint EM D-F$_1$ R-L DR | WikiAsp UniEval E-F$_1$ R-L |
|---|---|---|---|---|
| No retrieval | 72.9 | 33.8 24.2 33.3 28.4 | 40.1 32.5 36.4 34.4 | 47.1 14.1 26.4 |
| Single-time retrieval | 68.6 | 40.0 27.1 34.0 30.4 | 43.2 34.8 37.4 36.0 | 52.4 17.4 26.9 |
| *Multi-time retrieval* | | | | |
| Previous-window | 71.2 | 39.9 27.0 **34.3** 30.4 | 43.7 35.7 37.5 36.6 | 51.8 18.1 27.3 |
| Previous-sentence | 71.0 | 39.9 27.9 **34.3** 30.9 | 44.7 35.9 37.5 36.7 | 52.6 17.8 27.2 |
| FLARE (ours) | **77.3** | **41.3 28.2** 34.3 **31.1** | **46.2 36.7 37.7 37.2** | **53.4 18.9 27.6** |

Table 2: Comparison between FLARE and baselines on StrategyQA, ASQA, ASQA-hint, and WikiAsp. D-F$_1$ is Disambig-F$_1$, R-L is ROUGE-L, and E-F$_1$ is named entity-based F$_1$.

| | 2WikiMultihopQA EM F$_1$ Prec. Rec. | ASQA-hint EM D-F$_1$ R-L DR |
|---|---|---|
| Previous | 39.0 49.2 48.9 51.8 | 42.5 34.1 36.9 35.5 |
| Next | 48.8 57.6 57.1 60.5 | 45.9 35.7 37.5 36.6 |

Table 3: A head-to-head comparison between using the previous sentence and the next sentence for retrieval.

| #Tokens | EM | F$_1$ | Prec. | Rec. |
|---|---|---|---|---|
| 16 | 43.2 | 52.3 | 51.7 | 54.5 |
| 32 | 43.6 | 52.4 | 52.0 | 55.0 |
| 48 | 40.0 | 49.3 | 49.0 | 52.0 |
| All | 39.0 | 48.5 | 48.2 | 51.1 |

Table 4: Previous-window approaches using different numbers of tokens as queries.

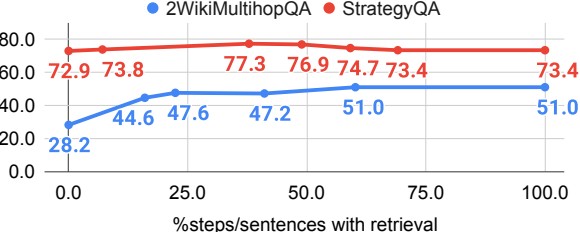

Figure 5: Performance (EM) of FLARE with respect to the percentage of steps/sentences with retrieval on 2WikiMultihopQA and StrategyQA.

ous window underperforms single-time retrieval on ASQA, which we hypothesize is because the previous window does not accurately reflect future intent. Since we focus on evaluating factuality, metrics with an emphasis on factual content (such as EM, Disambig-F$_1$, UniEval) are more reliable than metrics computed over all tokens (ROUGE-L).

## 6.2 Ablation Study

**Importance of forward-looking retrieval.** We first validate that forward-looking retrieval is more effective than past-context-based retrieval. We run ablation experiments on 2WikiMultihopQA and ASQA-hint comparing retrieval using the previous versus the next sentence. Specifically, both methods retrieve every sentence and directly use the complete previous/next sentence as queries. As shown in Table 3, using the next sentence to retrieve is clearly better than using the previous sentence, confirming our hypothesis.

We also run previous-window approaches using different numbers of past tokens as queries. As shown in Table 4, using too many tokens ($> 32$) in

the past hurts the performance, further confirming our hypothesis that previous context might not be relevant to intent of future generations.

**Importance of active retrieval.** Next, we investigate how active retrieval threshold $\theta$ affects performance. To alter our method from not retrieving to retrieving every sentence, we adjust the confidence threshold $\theta$ that determines when to trigger retrieval from 0 to 1. We then calculate the proportion of steps/sentences where retrieval is activated, and present the performance based on it. As shown in Figure 5, on 2WikiMultihopQA, the performance plateaus when the retrieval percentage exceeds 60%, indicating that retrieval when LMs are confident is not necessary. On StrategyQA, the performance drops when the retrieval percentage exceeds 50%, indicating that unnecessary retrieval can introduce noise and impede the original generation process. We found triggering retrieval for 40%-80% of sentences usually leads to a good performance across tasks/datasets.

**Effectiveness of different query formulation methods** We study implicit query formation by masking and explicit query formulation through question generation. In Table 5, we compare the performance of FLARE with different masking

| $\beta$ | EM | $F_1$ | Prec. | Rec. |
|---|---|---|---|---|
| 0.0 | 0.488 | 0.576 | 0.571 | 0.605 |
| 0.2 | 0.498 | 0.588 | 0.582 | 0.616 |
| 0.4 | 0.510 | 0.597 | 0.591 | 0.627 |
| 0.6 | 0.506 | 0.593 | 0.586 | 0.622 |

Table 5: Performance of FLARE with respect to the masking threshold $\beta$ on 2WikiMultihopQA.

| | ASQA-hint | | | | WikiAsp | | |
|---|---|---|---|---|---|---|---|
| | EM | D-$F_1$ | R-L | DR | UniEval | E-$F_1$ | R-L |
| Implicit | 45.7 | 36.9 | 37.7 | 37.3 | 53.4 | 18.8 | 27.7 |
| Explicit | 46.2 | 36.7 | 37.7 | 37.2 | 53.4 | 18.9 | 27.6 |

Table 6: A comparison between implicit and explicit query formulation methods in FLARE.

thresholds $\beta$. Retrieving directly with the complete sentence ($\beta = 0$) is worse than masking tokens with low probabilities, confirming our hypothesis that low-confidence erroneous tokens can distract retrievers. We compare implicit and explicit query formulation methods in Table 6. Performances of both methods are similar, indicating that both methods can effectively reflect information needs.

## 7   Related Work

We refer to subsection 2.2 and section 4 for extensively discussion on single-time and multi-time retrieval augmented LMs, which is the most relevant area to this paper.

**Iterative and adaptive retrieval**  Iterative retrieval and refinement has been studied in both text and code generation tasks (Peng et al., 2023; Zhang et al., 2023; Zemlyanskiy et al., 2022; Yu et al., 2023). FLARE differs from these methods in the granularity of generation and retrieval strategies. Adaptive retrieval has been studied in single-time retrieval scenarios based on either question popularity or generation probabilities (Mallen et al., 2022; Li et al., 2023), while we focus on long-form generation requiring active information access.

**Browser-enhanced LMs**  WebGPT (Nakano et al., 2021) and WebCPM (Qin et al., 2023) train LMs to interact with browser to enhance factuality using reinforcement learning or supervised training where multiple queries can be triggered before generation. FLARE is built on text-based retrievers but can be combined with a browser to potentially improve retrieval quality.

## 8   Conclusion

To aid long-form generation with retrieval augmentation, we propose an active retrieval augmented generation framework that decides when and what to retrieve during generation. We implement this framework with forward-looking active retrieval that iteratively uses the upcoming sentence to retrieve relevant information if it contains low-confidence tokens and regenerates the next sentence. Experimental results on 4 tasks/datasets demonstrate the effectiveness of our methods. Future directions include better strategies for active retrieval and developing efficient LM architectures for active information integration.

## 9   Limitations

We also conduct experiments on Wizard of Wikipedia (Dinan et al., 2019) and ELI5 (Fan et al., 2019), and found that FLARE did not provide significant gains. Wizard of Wikipedia is a knowledge-intensive dialogue generation dataset where the output is relatively short (~20 tokens on average) so retrieving multiple disparate pieces of information might not be necessary. ELI5 (Fan et al., 2019) is a long-form QA dataset requiring in-depth answers to open-ended questions. Due to issues mentioned in Krishna et al. (2021) such as difficulties of grounding generation in retrieval and evaluation, both single-time retrieval and FLARE did not provide significant gains over not using retrieval. From an engineering perspective, interleaving generation and retrieval with a naive implementation increases both overheads and the cost of generation. LMs need to be activated multiple times (once for each retrieval) and a caching-free implementation also requires recomputing the previous activation each time after retrieval. This issue can be potentially alleviated with special architectural designs that encode the retrieved documents $\mathcal{D}_{q_t}$ and the input/generation ($x/y_{<t}$) independently.

## Acknowledgements

This work was supported in part by a grant from the Singapore Defence Science and Technology Agency and the IBM PhD Fellowship. We thank Chunting Zhou, Amanda Bertsch, Uri Alon, Hiroaki Hayashi, Harsh Trivedi, Patrick Lewis, Timo Schick, Kaixin Ma, Shuyan Zhou, and Songwei Ge for their insightful discussions and help with the experiments.

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

## A FLARE Implementation Details

**FLARE$_{instruct}$ implementation details** We found that LMs can effectively combine retrieval and downstream task-related skills and generate meaningful search queries while performing the task. However, there are two issues: (1) LMs tend to generate fewer search queries than necessary. (2) Generating excessive search queries can disrupt answer generation and adversely affect performance. We address these issues using two methods respectively. First, we increase the logit of the token "[" by 2.0 to improve the chances of LMs generating "[Search(query)]". Second, whenever LMs generate a search query, we use it to retrieve relevant information, promptly remove it from the generation, and generate the next few tokens while forbidding "[" by adding a large negative value to the logit of "[".

**The initial query of FLARE.** FLARE starts with the user input $x$ as the initial query to retrieve documents to generate the first sentence $\hat{s}_1 = \text{LM}([\mathcal{D}_x, x])$ to bootstrap the iterative generation process. For the following steps, the temporary forward-looking sentence is generated without retrieved documents.

**Sentence tokenization.** For each step $t$, we generate 64 tokens which are longer than most sentences, and use NLTK sentence tokenizer[5] to extract the first sentence and discard the rest.

**Efficiency** As shown in subsection 6.2, on average retrieval is triggered for $30\% \sim 60\%$ of sentences depending on downstream tasks. In comparision, KNN-LM (Khandelwal et al., 2020) retrieves every token, RETRO or IC-RALM (Borgeaud et al., 2022; Ram et al., 2023) retrievers every $4 \sim 32$ tokens, and IRCoT (Trivedi et al., 2022) retrieves every sentence. Compared to single-time retrieval, however, interleaving retrieval and generation with a naive implementation indeed increases overheads, which we discuss in the limitation section (section 9).

## B Datasets and Settings

Datasets, metrics, and experimental settings are summarized in Table 7.

---

[5]https://www.nltk.org/api/nltk.tokenize.PunktSentenceTokenizer.html

**Multihop QA** For "Why did the founder of Versus die?", the output we aim to generate is "The founder of Versus was Gianni Versace. Gianni Versace was shot and killed on the steps of his Miami Beach mansion on July 15, 1997. So the answer is shot." We use 8 exemplars from Trivedi et al. (2022) listed in Prompt D.4 for in-context learning, BM25 as the retriever, and Wikipedia articles as the retrieval corpus. Similar to the observation in Trivedi et al. (2022), we found incorporating retrieval results for exemplars improves the performance, we use the input $x$ of each exemplar to retrieve several documents and then add them using the format in Prompt D.1. We found increasing the number of retrieval documents often increases performance. Therefore, we use the maximum number of documents that can fit within the input length limit of `text-davinci-003`, which is 2 for 2Wiki-MultihopQA.

**Commonsense Reasoning** For "Would a pear sink in water?", the output we aim to generate is "The density of a pear is about 0.6g/cm$^3$, which is less than water. Objects less dense than water float. Thus, a pear would float. So the final answer is no." We use 6 exemplars from Wei et al. (2022) listed in Prompt D.5, BM25 on the Wikipedia corpus, and 3 retrieved documents to run experiments.

**Long-form QA** For "Where do the Philadelphia Eagles play their home games?", the output we aim to generate is "We need to consider the different possible locations or venues that could be considered the home field of the Philadelphia Eagles. These include the city, the sports complex, or the stadium. Therefore, this question has 3 interpretations and the answers are: (1) The city is Philadelphia. (2) The sports complex is the South Philadelphia Sports Complex. (3) The stadium is the Lincoln Financial Field stadium." For both the original setting (ASQA) and the setting with hints (ASQA-hint), we manually annotate 8 exemplars (Prompt D.6 and D.8), use BM25 on the Wikipedia corpus, and 3 retrieved documents to run experiments.

**Open-domain Summarization** The original WikiAsp dataset is designed for multi-document summarization and provides a list of references to systems. We converted it into the open-domain setting by removing the associated references and instead gathering information from the open web. For "Generate a summary about Echo School (Ore-

gon) including the following aspects: academics, history.", the output we aim to generate is "# Academics. In 2008, 91% of the school's seniors received their high school diploma... # History. The class of 2008 was the 100th class in the school's history." where # is used to indicate aspects. We manually annotate 4 exemplars (Prompt D.10), and use the Bing search engine to retrieve 5 documents from the open web. To avoid leaking, we exclude several Wikipedia-related domains listed in Table 8 from Bing's search results.

## C  Hyperparameters

Hyperparameters of FLARE on different datasets are listed in Table 9.

## D  Prompts and Few-shot exemplars

The prompt used to linearize multiple documents is shown in Prompt D.1. The prompt used in self-ask (Press et al., 2022) is shown in Prompt D.2. Prompts and exemplars of different tasks/datasets are shown in Prompt D.3, D.4, D.5, D.6, D.8, and D.10, respectively.

---

**Prompt D.1: document formatting**

Search results:
[1] Document 1
[2] Document 2
...
The user input $x$

---

**Prompt D.2: multihop QA with self-ask**

Question: Who lived longer, Theodor Haecker or Harry Vaughan Watkins?
Are follow up questions needed here: Yes.
Follow up: How old was Theodor Haecker when he died?
Intermediate answer: Theodor Haecker was 65 years old when he died.
Follow up: How old was Harry Vaughan Watkins when he died?
Intermediate answer: Harry Vaughan Watkins was 69 years old when he died.
So the final answer is: Harry Vaughan Watkins.

| Settings | 2WikiMultihopQA (Ho et al., 2020) | StrategyQA (Geva et al., 2021) | ASQA (Stelmakh et al., 2022) | WikiAsp (Hayashi et al., 2021) |
|---|---|---|---|---|
| *Dataset statistics* | | | | |
| Task | multihop QA | commonsense QA | long-form QA | open-domain summarization |
| #Examples | 500 | 229 | 500 | 500 |
| *Evaluation settings* | | | | |
| Metrics | EM, $F_1$, Prec., Rec. | EM | EM, Disambig-$F_1$, ROUGE, DR | UniEval, entity-$F_1$, ROUGE |
| *Retrieval settings* | | | | |
| Corpus | Wikipedia | Wikipedia | Wikipedia | open web |
| Retriever | BM25 | BM25 | BM25 | Bing |
| Top-k | 2 | 3 | 3 | 5 |
| *Prompt format* | | | | |
| #Exemplars | 8 | 6 | 8 | 4 |
| Ret. for exemplars | ✓ | ✗ | ✗ | ✗ |

Table 7: Dataset statistics and experimental settings of different tasks.

wikipedia.org, wikiwand.com, wiki2.org, wikimedia.org

Table 8: Wikipedia-related domains excluded from Bing's search results.

| Dataset | $\theta$ | $\beta$ | Query formulation | Combine single- & multi-time retrieval |
|---|---|---|---|---|
| 2WikiMultihopQA | 0.8 | 0.4 | implicit | ✗ |
| StrategyQA | 0.4 | 0.4 | implicit | ✗ |
| ASQA & ASQA-hint | 0.8 | 0.4 | explicit | ✓ |
| WikiAsp | 0.8 | 0.4 | explicit | ✓ |

Table 9: Hyperparameters of FLARE on different datasets.

## Prompt D.3: retrieval instructions for 2WikiMultihopQA

Skill 1. Use the Search API to look up relevant information by writing "[Search(term)]" where "term" is the search term you want to look up. For example:

Question: But what are the risks during production of nanomaterials?
Answer (with Search): [Search(nanomaterial production risks)] Some nanomaterials may give rise to various kinds of lung damage.

Question: The colors on the flag of Ghana have the following meanings.
Answer (with Search): Red is for [Search(Ghana flag red meaning)] the blood of martyrs, green for forests, and gold for mineral wealth.

Question: Metformin is the first-line drug for what?
Answer (with Search): [Search(Metformin first-line drug)] patients with type 2 diabetes and obesity.

Skill 2. Answer questions by thinking step-by-step. First, write out the reasoning steps, then draw the conclusion. For example:

Question: When did the director of film Hypocrite (Film) die?
Answer (with step-by-step): The film Hypocrite was directed by Miguel Morayta. Miguel Morayta died on 19 June 2013. So the answer is 19 June 2013.

Question: Are both Kurram Garhi and Trojkrsti located in the same country?
Answer (with step-by-step): Kurram Garhi is located in the country of Pakistan. Trojkrsti is located in the country of Republic of Macedonia. Thus, they are not in the same country. So the answer is no.

Question: Do director of film Coolie No. 1 (1995 Film) and director of film The Sensational Trial have the same nationality?
Answer (with step-by-step): Coolie No. 1 (1995 film) was directed by David Dhawan. The Sensational Trial was directed by Karl Freund. David Dhawan's nationality is India. Karl Freund's nationality is Germany. Thus, they do not have the same nationality. So the answer is no.

Question: Who is Boraqchin (Wife Of Ögedei)'s father-in-law?
Answer (with step-by-step): Boraqchin is married to Ögedei Khan. Ögedei Khan's father is Genghis Khan. Thus, Boraqchin's father-in-law is Genghis Khan. So the answer is Genghis Khan.

Question: Who was born first out of Martin Hodge and Ivania Martinich?
Answer (with step-by-step): Martin Hodge was born on 4 February 1959. Ivania Martinich was born on 25 July 1995. Thus, Martin Hodge was born first. So the answer is Martin Hodge.

Question: When did the director of film Laughter In Hell die?
Answer (with step-by-step): The film Laughter In Hell was directed by Edward L. Cahn. Edward L. Cahn died on August 25, 1963. So the answer is August 25, 1963.

Question: Which film has the director died later, The Gal Who Took the West or Twenty Plus Two?
Answer (with step-by-step): The film Twenty Plus Two was directed by Joseph M. Newman. The Gal Who Took the West was directed by Frederick de Cordova. Joseph M. Newman died on January 23, 2006. Fred de Cordova died on September 15, 2001. Thus, the person to die later from the two is Twenty Plus Two. So the answer is Twenty Plus Two.

Question: Who is the grandchild of Krishna Shah (Nepalese Royal)?
Answer (with step-by-step): Krishna Shah has a child named Rudra Shah. Rudra Shah has a child named Prithvipati Shah. Thus, Krishna Shah has a grandchild named Prithvipati Shah. So the answer is Prithvipati Shah.

Now, combine the aforementioned two skills. First, write out the reasoning steps, then draw the conclusion, where the reasoning steps should also utilize the Search API "[Search(term)]" whenever possible.

Question: Where did Minbyauk Thihapate's wife die?
Answer (with step-by-step & Search):

## Prompt D.4: exemplars of 2WikiMultihopQA

Question: When did the director of film Hypocrite (Film) die?
Answer: The film Hypocrite was directed by Miguel Morayta. Miguel Morayta died on 19 June 2013. So the answer is 19 June 2013.

Question: Are both Kurram Garhi and Trojkrsti located in the same country?
Answer: Kurram Garhi is located in the country of Pakistan. Trojkrsti is located in the country of Republic of Macedonia. Thus, they are not in the same country. So the answer is no.

Question: Do director of film Coolie No. 1 (1995 Film) and director of film The Sensational Trial have the same nationality?
Answer: Coolie No. 1 (1995 film) was directed by David Dhawan. The Sensational Trial was directed by Karl Freund. David Dhawan's nationality is India. Karl Freund's nationality is Germany. Thus, they do not have the same nationality. So the answer is no.

Question: Who is Boraqchin (Wife Of Ögedei)'s father-in-law?
Answer: Boraqchin is married to Ögedei Khan. Ögedei Khan's father is Genghis Khan. Thus, Boraqchin's father-in-law is Genghis Khan. So the answer is Genghis Khan.

Question: Who was born first out of Martin Hodge and Ivania Martinich?
Answer: Martin Hodge was born on 4 February 1959. Ivania Martinich was born on 25 July 1995. Thus, Martin Hodge was born first. So the answer is Martin Hodge.

Question: When did the director of film Laughter In Hell die?
Answer: The film Laughter In Hell was directed by Edward L. Cahn. Edward L. Cahn died on August 25, 1963. So the answer is August 25, 1963.

Question: Which film has the director died later, The Gal Who Took the West or Twenty Plus Two?
Answer: The film Twenty Plus Two was directed by Joseph M. Newman. The Gal Who Took the West was directed by Frederick de Cordova. Joseph M. Newman died on January 23, 2006. Fred de Cordova died on September 15, 2001. Thus, the person to die later from the two is Twenty Plus Two. So the answer is Twenty Plus Two.

Question: Who is the grandchild of Krishna Shah (Nepalese Royal)?
Answer: Krishna Shah has a child named Rudra Shah. Rudra Shah has a child named Prithvipati Shah. Thus, Krishna Shah has a grandchild named Prithvipati Shah. So the answer is Prithvipati Shah.

Question: Which country the director of film Citizen Mavzik is from?
Answer:

## Prompt D.5: exemplars of StrategyQA

Generate a yes or no answer to the following question.
Question: Do hamsters provide food for any animals?
Answer: Hamsters are prey animals. Prey are food for predators. Thus, hamsters provide food for some animals. So the final answer is yes.

Generate a yes or no answer to the following question.
Question: Could Brooke Shields succeed at University of Pennsylvania?
Answer: Brooke Shields went to Princeton University. Princeton University is about as academically rigorous as the University of Pennsylvania. Thus, Brooke Shields could also succeed at the University of Pennsylvania. So the final answer is yes.

Generate a yes or no answer to the following question.
Question: Hydrogen's atomic number squared exceeds number of Spice Girls?
Answer: Hydrogen has an atomic number of 1. 1 squared is 1. There are 5 Spice Girls. Thus, Hydrogen's atomic number squared is less than 5. So the final answer is no.

Generate a yes or no answer to the following question.
Question: Is it common to see frost during some college commencements?
Answer: College commencement ceremonies can happen in December, May, and June. December is in the winter, so there can be frost. Thus, there could be frost at some commencements. So the final answer is yes.

Generate a yes or no answer to the following question.
Question: Could a llama birth twice during War in Vietnam (1945-46)?
Answer: The War in Vietnam was 6 months. The gestation period for a llama is 11 months, which is more than 6 months. Thus, a llama could not give birth twice during the War in Vietnam. So the final answer is no.

Generate a yes or no answer to the following question.
Question: Would a pear sink in water?
Answer: The density of a pear is about 0.6g/cm$^3$, which is less than water. Objects less dense than water float. Thus, a pear would float. So the final answer is no.

Generate a yes or no answer to the following question.
Question: Did Spiderman fight against Falcon in the MCU?
Answer:

## Prompt D.6: exemplars of ASQA

Given an ambiguous question, figure out its interpretations and answer them one by one.
Question: Who played bonnie in gone with the wind?
Answer: This question is ambiguous in terms of which version or adaptation of Gone with the Wind is being referred to. In order to figure out its interpretations, we need to consider different versions or adaptations of Gone with the Wind. Gone with the Wind has two versions or adaptations: the 1939 film Gone with the Wind or the 2008 musical Gone with the Wind. Therefore, this question has 2 interpretations: (1) Who played Bonnie in the 1939 film Gone with the Wind? (2) Who played Bonnie in the 2008 musical Gone with the Wind? The answers to all interpretations are: (1) The 1939 film Gone with the Wind's character Bonnie was played by Eleanore Cammack "Cammie" King. (2) The 2008 musical Gone with the Wind's character Bonnie was played by Leilah de Meza.

Given an ambiguous question, figure out its interpretations and answer them one by one.
Question: What is the second largest city in the usa?
Answer: This question is ambiguous in terms of the criteria being used to determine the second largest city in the USA. In order to figure out its interpretations, we need to consider different criteria to determine a city's size. City size can be measured by two criteria: population or area. Therefore, this question has 2 interpretations: (1) What is the second largest city in the USA by population? (2) What is the second largest city in the USA by area? The answers to all interpretations are: (1) The second largest city in the USA by population is Los Angeles, California. (2) The second largest city in the USA by area is Juneau, Alaska.

Given an ambiguous question, figure out its interpretations and answer them one by one.
Question: When was bohemian rhapsody released as a single?
Answer: This question is ambiguous in terms of which country's release of the single is being referred to. In order to figure out its interpretations, we need to consider different countries where Bohemian Rhapsody is released. Bohemian Rhapsody was released in the United Kingdom and in the United States on different dates. Therefore, this question has 2 interpretations: (1) When was Bohemian Rhapsody released as a single in the United Kingdom? (2) When was Bohemian Rhapsody released as a single in the United States? The answers to all interpretations are: (1) Bohemian Rhapsody was released as a single in the United Kingdom on 31 October 1975. (2) Bohemian Rhapsody was released as a single in the United States on December 1975.

Given an ambiguous question, figure out its interpretations and answer them one by one.
Question: Where do the philadelphia eagles play their home games?
Answer: This question is ambiguous in terms of which specific location or venue is being referred to. In order to figure out its interpretations, we need to consider the different possible locations or venues that could be considered the home field of the Philadelphia Eagles. These include the city, the sports complex, or the stadium. Therefore, this question has 3 interpretations: (1) What city do the Philadelphia Eagles play their home games? (2) In what sports complex do the Philadelphia Eagles play their home games? (3) What stadium do the Philadelphia Eagles play their home games? The answers to all interpretations are: (1) Philadelphia Eagles play their home games in the city Philadelphia. (2) Philadelphia Eagles play their home games in the South Philadelphia Sports Complex. (3) Philadelphia Eagles play their home games in the Lincoln Financial Field stadium.

Given an ambiguous question, figure out its interpretations and answer them one by one.
Question: When did xbox one come out in australia?
Answer: This question is ambiguous in terms of which specific version of the Xbox One is being referred to. In order to figure out its interpretations, we need to consider the different versions of the Xbox One that have been released. Xbox One has two versions: the Xbox One video game console or the Xbox One X high-end model. Therefore, this question has 2 interpretations: (1) When did the Xbox One release in Australia? (2) When did the Xbox One X release in Australia? The answers to all interpretations are: (1) The Xbox One video game console was released in Australia on November 22, 2013. (2) The Xbox One X video game console was released in Australia on November 7, 2017.

Given an ambiguous question, figure out its interpretations and answer them one by one.
Question: When does the movie summer of 84 come out?
Answer: This question is ambiguous in terms of which release of the movie is being referred to. In order to figure out its interpretations, we need to consider different releases of the movie Summer of '84. The movie Summer of '84 is first released at the Sundance Festival before it's released throughout the US. Therefore, this question has 2 interpretations: (1) When did the movie Summer of '84 first release at the Sundance Festival? (2) When did the movie Summer of '84 first release throughout the US? The answers to all interpretations are: (1) Summer of '84 was released at the Sundance Festival on January 22, 2018. (2) Summer of '84 was released throughout the US on August 10, 2018.

## Prompt D.7: exemplars of ASQA (cont.)

Given an ambiguous question, figure out its interpretations and answer them one by one.
Question: What was roy orbison's first number one hit?
Answer: This question is ambiguous in terms of which specific chart or region is being referred to. In order to figure out its interpretations, we need to consider the different charts and regions where Roy Orbison's music was popular. Roy Orbison is popular in both the US Hot 100 and Canada, and the UK and Ireland. Therefore, this question has 2 interpretations: (1) What was Roy Orbison's first number one hit in the US Hot 100 and Canada? (2) What was Roy Orbison's first number one hit in the UK and Ireland? The answers to all interpretations are: (1) Running Scared was the first number one hit for Roy Orbison in the US Hot 100 and Canada. (2) Only the Lonely (Know the Way I Feel) was the first number one hit for Roy Orbison in the UK and Ireland.

Given an ambiguous question, figure out its interpretations and answer them one by one.
Question: What is the criminal's name in the breakfast club?
Answer: This question is ambiguous in terms of which specific name is being referred to - the character's name or the actor's name. In order to figure out its interpretations, we need to consider both possibilities: the character's name or the actor's name. Therefore, this question has 2 interpretations: (1) What is the criminal's character name in The Breakfast Club? (2) What is the the name of the actor who played the criminal in The Breakfast Club? The answers to all interpretations are: (1) John Bender was the name of the criminal's character in The Breakfast Club. (2) Judd Nelson was the actor of the criminal in The Breakfast Club.

Given an ambiguous question, figure out its interpretations and answer them one by one.
Question: How many state parks are there in virginia?
Answer:

## Prompt D.8: exemplars of ASQA-hint

Given an ambiguous question and a hint on which aspect of the question is ambiguous, figure out its interpretations and answer them one by one.
Question: Who played bonnie in gone with the wind?
Hint: This question is ambiguous in terms of which version or adaptation of Gone with the Wind is being referred to.
Answer: In order to figure out its interpretations, we need to consider different versions or adaptations of Gone with the Wind. Gone with the Wind has two versions or adaptations: the 1939 film Gone with the Wind or the 2008 musical Gone with the Wind. Therefore, this question has 2 interpretations: (1) Who played Bonnie in the 1939 film Gone with the Wind? (2) Who played Bonnie in the 2008 musical Gone with the Wind? The answers to all interpretations are: (1) The 1939 film Gone with the Wind's character Bonnie was played by Eleanore Cammack "Cammie" King. (2) The 2008 musical Gone with the Wind's character Bonnie was played by Leilah de Meza.

Given an ambiguous question and a hint on which aspect of the question is ambiguous, figure out its interpretations and answer them one by one.
Question: What is the second largest city in the usa?
Hint: This question is ambiguous in terms of the criteria being used to determine the second largest city in the USA.
Answer: In order to figure out its interpretations, we need to consider different criteria to determine a city's size. City size can be measured by two criteria: population or area. Therefore, this question has 2 interpretations: (1) What is the second largest city in the USA by population? (2) What is the second largest city in the USA by area? The answers to all interpretations are: (1) The second largest city in the USA by population is Los Angeles, California. (2) The second largest city in the USA by area is Juneau, Alaska.

Given an ambiguous question and a hint on which aspect of the question is ambiguous, figure out its interpretations and answer them one by one.
Question: When was bohemian rhapsody released as a single?
Hint: This question is ambiguous in terms of which country's release of the single is being referred to.
Answer: In order to figure out its interpretations, we need to consider different countries where Bohemian Rhapsody is released. Bohemian Rhapsody was released in the United Kingdom and in the United States on different dates. Therefore, this question has 2 interpretations: (1) When was Bohemian Rhapsody released as a single in the United Kingdom? (2) When was Bohemian Rhapsody released as a single in the United States? The answers to all interpretations are: (1) Bohemian Rhapsody was released as a single in the United Kingdom on 31 October 1975. (2) Bohemian Rhapsody was released as a single in the United States on December 1975.

Given an ambiguous question and a hint on which aspect of the question is ambiguous, figure out its interpretations and answer them one by one.
Question: Where do the philadelphia eagles play their home games?
Hint: This question is ambiguous in terms of which specific location or venue is being referred to.
Answer: In order to figure out its interpretations, we need to consider the different possible locations or venues that could be considered the home field of the Philadelphia Eagles. These include the city, the sports complex, or the stadium. Therefore, this question has 3 interpretations: (1) What city do the Philadelphia Eagles play their home games? (2) In what sports complex do the Philadelphia Eagles play their home games? (3) What stadium do the Philadelphia Eagles play their home games? The answers to all interpretations are: (1) Philadelphia Eagles play their home games in the city Philadelphia. (2) Philadelphia Eagles play their home games in the South Philadelphia Sports Complex. (3) Philadelphia Eagles play their home games in the Lincoln Financial Field stadium.

Given an ambiguous question and a hint on which aspect of the question is ambiguous, figure out its interpretations and answer them one by one.
Question: When did xbox one come out in australia?
Hint: This question is ambiguous in terms of which specific version of the Xbox One is being referred to.
Answer: In order to figure out its interpretations, we need to consider the different versions of the Xbox One that have been released. Xbox One has two versions: the Xbox One video game console or the Xbox One X high-end model. Therefore, this question has 2 interpretations: (1) When did the Xbox One release in Australia? (2) When did the Xbox One X release in Australia? The answers to all interpretations are: (1) The Xbox One video game console was released in Australia on November 22, 2013. (2) The Xbox One X video game console was released in Australia on November 7, 2017.

Given an ambiguous question and a hint on which aspect of the question is ambiguous, figure out its interpretations and answer them one by one.
Question: When does the movie summer of 84 come out?
Hint: This question is ambiguous in terms of which release of the movie is being referred to.
Answer: In order to figure out its interpretations, we need to consider different releases of the movie Summer of '84. The movie Summer of '84 is first released at the Sundance Festival before it's released throughout the US. Therefore, this question has 2 interpretations: (1) When did the movie Summer of '84 first release at the Sundance Festival? (2) When did the movie Summer of '84 first release throughout the US? The answers to all interpretations are: (1) Summer of '84 was released at the Sundance Festival on January 22, 2018. (2) Summer of '84 was released throughout the US on August 10, 2018.

Given an ambiguous question and a hint on which aspect of the question is ambiguous, figure out its interpretations and answer them one by one.
Question: What was roy orbison's first number one hit?
Hint: This question is ambiguous in terms of which specific chart or region is being referred to.
Answer: In order to figure out its interpretations, we need to consider the different charts and regions where Roy Orbison's music was popular. Roy Orbison is popular in both the US Hot 100 and Canada, and the UK and Ireland. Therefore, this question has 2 interpretations: (1) What was Roy Orbison's first number one hit in the US Hot 100 and Canada? (2) What was Roy Orbison's first number one hit in the UK and Ireland? The answers to all interpretations are: (1) Running Scared was the first number one hit for Roy Orbison in the US Hot 100 and Canada. (2) Only the Lonely (Know the Way I Feel) was the first number one hit for Roy Orbison in the UK and Ireland.

Given an ambiguous question and a hint on which aspect of the question is ambiguous, figure out its interpretations and answer them one by one.
Question: What is the criminal's name in the breakfast club?
Hint: This question is ambiguous in terms of which specific name is being referred to - the character's name or the actor's name.
Answer: In order to figure out its interpretations, we need to consider both possibilities: the character's name or the actor's name. Therefore, this question has 2 interpretations: (1) What is the criminal's character name in The Breakfast Club? (2) What is the the name of the actor who played the criminal in The Breakfast Club? The answers to all interpretations are: (1) John Bender was the name of the criminal's character in The Breakfast Club. (2) Judd Nelson was the actor of the criminal in The Breakfast Club.

Given an ambiguous question and a hint on which aspect of the question is ambiguous, figure out its interpretations and answer them one by one.
Question: How many state parks are there in virginia?
Hint: This question is ambiguous in terms of the time frame or period being referred to.
Answer:

## Prompt D.10: exemplars of WikiAsp

Generate a summary about Aslanhane Mosque including the following aspects: location, history with one aspect per line.
# Location
The mosque is in the old quarter of ankara next to ankara castle. With an altitude of 947 metres (3,107 ft) it overlooks ankara at 39°56'12"N 32°51'55"E.
# History
The mosque is one of the oldest mosques in Turkey still standing. It was built during the reign of Mesud II of the Anatolian Seljuks in 1290. Its architect was Ebubekir Mehmet. It was commissioned by two Ahi leaders named Hüsamettin and Hasaneddin. However, in 1330, it was repaired by another Ahi leader named Şerafettin after whom the mosque was named. After several minor repairs the mosque was restored by the directorate general of foundations in 2010-2013 term.

Generate a summary about Untold Legends: The Warrior's Code including the following aspects: reception, gameplay, development with one aspect per line.
# Reception
The game received "mixed or average reviews" according to video game review aggregator Metacritic.
# Gameplay
The warrior's code is a hack n' slash action role-playing game, which concentrates on action-oriented combat.
# Development
As a pre-order bonus, the game was shipped with a small action figure of the Guardian class.

Generate a summary about Raid on St. Augustine including the following aspects: aftermath, background with one aspect per line.
# Aftermath
Once the English had gone Menéndez and the rest of the Spanish settlers returned to find a smoldering ruins and very little left. He soon and begged for help from the viceroy of Cuba and the settlement took a while to build itself back up. The destroyed fort was replaced with the present day Castillo de San Marcos.
# Background
War had already been unofficially declared by Philip II of Spain after the Treaty of Nonsuch in which Elizabeth I had offered her support to the rebellious Protestant Dutch rebels. The Queen through Francis Walsingham ordered Sir Francis Drake to lead an expedition to attack the Spanish New World in a kind of preemptive strike. Sailing from Plymouth, England, he struck first at Santiago in November 1585 then across the Atlantic at the Spanish new world city of Santo Domingo of which was captured and ransomed on 1 January 1586 and following that successfully attacked the important city of Cartagena on 19 February. Drake wanted to strike at another Spanish city on the Main before finally visiting and replenishing Sir Walter Raleigh's new colony of Roanoke Colony on the American East Coast. Then after this he hoped to make the Transatlantic crossing back to England. The fleet headed north, and in late April Drake put into the Spanish Cuban mainland and his men dug wells in search of fresh water and gathered supplies to help counter an outbreak of dysentery after which he moved on. The fleet traveled north within sight of land on the Florida peninsula sailing past the West coast. On 27 May 1586 as they approached further north a small fort was spotted on the shore, with a small inlet close by. This was the location of St Augustine, the most northerly town in Spain's New World Empire, and the oldest permanent colonial settlement in North America. Drake knew of the place and was also aware of the fact that the spanish under Pedro Menéndez de Avilés had ordered all of the French Huguenot colonists that had tried to settle in the area executed. Drake decided on one final opportunity to raid and plunder, and a chance to avenge his fellow Protestants.

Generate a summary about Lakewood (Livingston, Alabama) including the following aspects: architecture, history with one aspect per line.
# Architecture
The house has a plan that is relatively rare in early Alabama architecture. The plan features a brick ground floor that is topped by one-and-a-half-stories of wood-frame construction. The ground floor originally contained domestic spaces, with the formal rooms on the principle floor and bedrooms on the upper floor. A central hallway is present on all levels. The facade is five bays wide, with central entrance doors on the ground and principle floors. The bays are divided by two-story Doric pilasters, with the middle third of the facade occupied by a two-tiered tetrastyle Doric portico. Two curved wrought iron staircases ascend from ground level to the front center of the upper portico, leading to the formal entrance.
# History
Lakewood was built for Joseph lake, a native of North Carolina, by Hiram W. Bardwell, a master builder. Construction was completed in 1840. Located adjacent to the University of West Alabama, Julia Strudwick Tutwiler, a Lake relative, periodically resided in the house from 1881 to 1910 while she served as president of the university. It was then known as Livingston Normal College. The house was extensively photographed by Alex Bush for the Historic American Buildings Survey in November and December 1936. Lakewood has continued to be owned by descendants of the Lake family to the current day. The house and its surviving 10 acres (4.0 ha) of grounds were listed on the Places in Peril in 2012 due to the immediate threat of its acquisition by developers.

Generate a summary about Carlos Moedas including the following aspects: biography, early life, political career with one aspect per line.