# OpenReview forum: "Active Retrieval Augmented Generation"
_EMNLP/2023/Conference — EMNLP 2023 Main_

### Official Review · Reviewer_6yUm · 2023-07-31

**Soundness:** 3

**Excitement:**

3: Ambivalent: It has merits (e.g., it reports state-of-the-art results, the idea is nice), but there are key weaknesses (e.g., it describes incremental work), and it can significantly benefit from another round of revision. However, I won't object to accepting it if my co-reviewers champion it.

**Missing References:**

N/A

**Paper Topic And Main Contributions:**

This paper proposes a novel approach, named Forward-Looking Active Retrieval Augmented generation (FLARE), to reduce the hallucination for long-form text generation. Specifically, they iteratively generate a temporary next sentence and check whether it contains low-probality tokens. If so, they retrieve again with the reformulated query and regenerate the sentence. FLARE can be used for any existing LMs during inference time without additional training. On four QA datasets (MultihopQA, StrategyQA, ASQA, WikiAsp), they demonstrate the effectiveness of FLARE over all baselines. However, according to the author, their approach did not provide significant gains on Wizard of Wikipedia and ELI5. Overall, this paper is well-written and brings insights to the community that forward active retrieval can be useful for QA tasks and reduce hallucination.

**Questions For The Authors:**

1. Can you report the latency and efficiency between your approach and baselines?
2. What's the threshold value (\beta and \theta) for different tasks? Do you choose it by empirical experiments on different tasks?


**Reasons To Accept:**

1. Their proposed approach FLARE demonstrate that actively decodes when and what to retrieve throughout the generation process can bring gains for multiple QA tasks.
2. They designed two approaches that mask sentences as implicit queries and generate questions as explicit queries for iteratively retrieval to overcome the errors that might exist in the next sentence.

**Reasons To Reject:**

1. The first approach FLARE with Retrieval Instruction is similar to Toolformer, while the second Direct FLARE that utilize the confidence to decide retrieve is straightforward and might have been explored. The author should do some existing literature.
2. The author should report the efficiency and latency since the proposed approach requires multiple iterative retrievals.

**Reproducibility:**

4: Could mostly reproduce the results, but there may be some variation because of sample variance or minor variations in their interpretation of the protocol or method.

**Reviewer Confidence:**

4: Quite sure. I tried to check the important points carefully. It's unlikely, though conceivable, that I missed something that should affect my ratings.

---

> ### Author Rebuttal · Authors · 2023-08-28
>
> Thanks for the constructive feedback!
>
> > Q1 (reason to reject 1): Novelty of FLARE-instruct and FLARE-direct.
>
> FLARE-instruct is indeed inspired by Toolformer (as mentioned in line 221) but one significant difference is that we aim to generate multiple queries in long-form generation scenarios. Regarding the novelty/contribution of this work, we view FLARE-direct as our major technical contribution and view FLARE-instruct as an exploratory variation/ablation. We discussed several existing works in the related work section (in the appendix E) and we will move this section to the main body in the revised version. As far as we know, FLARE-direct is the first method that actively decides when to retrieve with probabilities and what to retrieve with forward-looking. We are happy to cite and compare other baselines if the reviewer can provide relevant references.
>
> > Q2 (reason to reject 2): efficiency and latency of the proposed methods.
>
> Thanks for the suggestion! We agree that iterative retrieval during generation increases latency compared to single-time retrieval, which we mentioned in the “efficiency” paragraph in appendix A. Generation latency is correlated with the number of retrieval triggered during generation. Since our method actively decides when to retrieve based on probability, it is faster than other multi-time retrieval baselines that trigger retrieval regularly (every sentence/few tokens). We do plan to report latency, but since we access OpenAI LMs through API calls, the measured latency might be overestimated because of the overhead of API calling. One future direction is to implement FLARE on top of LLaMA, where we have full control of the generation process and the latency measurement is more accurate.
>
> > Q3 (question 2): threshold value (beta and theta) for different tasks.
>
> We choose these threshold values (reported in table 8 in the appendix) based on the performance on the development set.

---

### Official Review · Reviewer_uvL6 · 2023-08-04

**Soundness:** 3

**Excitement:**

4: Strong: This paper deepens the understanding of some phenomenon or lowers the barriers to an existing research direction.

**Paper Topic And Main Contributions:**

This paper focuses on two important questions of retrieval augmented large language models: when and what to retrieve. Different from the previous single-time retrieval and inflexible multi-time retrieval, the authors propose an active retrieval method to trigger retrieval at the right time. They also propose a future-aware retrieval approach to use temporary next sentences as queries to aid future generations. Empirical results show that FLARE outperforms baselines on all four datasets.


**Questions For The Authors:**

1. For strategyQA data set, why only 229 data were extracted for testing?
2. What is the length of each wikipedia documentation chunk？
3. Is there any treatment for the results returned by bing search, such as length truncation？


**Reasons To Accept:**

1. This paper proposes a generic retrieval augmented LLMs framework, which can actively decide when and what to retrieve across the course of the generation.
2. The experiment is thorough. The authors verified the effectiveness of the method on different types of tasks and made detailed experimental analysis.
3. The article is clear and easy to follow.

**Reasons To Reject:**

1. On StrategyQA, ASQA, ASQA-Hint, and WikiAsp tasks, the authors only compared FLARE to simple baselines, not to other baseline methods.

**Reproducibility:**

4: Could mostly reproduce the results, but there may be some variation because of sample variance or minor variations in their interpretation of the protocol or method.

**Reviewer Confidence:**

4: Quite sure. I tried to check the important points carefully. It's unlikely, though conceivable, that I missed something that should affect my ratings.

---

> ### Author Rebuttal · Authors · 2023-08-28
>
> Thanks for the constructive feedback, and we are happy that you appreciated the thorough experiments!
>
> > Q (reason to reject 1): baselines (previous-sentence and question decomposition) are not tested on the other 3 datasets.
>
> We do have the performance of the previous-sentence method on the other 3 dataset and it is slightly worse than the previous-window method, similar to the observation on 2WikiMultihopQA. For example, EM of the previous-sentence approach on StrategyQA/ASQA are 71.0/49.2. We will add these numbers in the revised version.
>
> The question decomposition method proposed in self-ask (Press et al., 2022), as the name suggests, specializes in and is proposed for solving compositional questions. They are questions that require multiple hops of querying and can straightforwardly break down into simpler questions. Therefore we reuse their manually designed prompts on multi-hop QA (i.e., 2WikiMultiHopQA) as a critical baseline. This helps us demonstrate FLARE works well compared to specialized prompting strategies. On the other hand, when it comes to the other datasets such as summarization, the way to decompose becomes unclear, and to the best of our knowledge, there is no such existing work. This is also why FLARE is a more versatile method that does not require special designs.
>
> > Q (question 1): 229 examples for strategyQA.
>
> We follow previous papers in using the development set of StrategyQA (10% of the full dataset) to run experiments.
>
> > Q (question 2): length of each wikipedia documentation chunk.
>
> We use the Wikipedia corpus released by DPR (Karpukhin et al., 2020) where each chunk has about 100 words.
>
> > Q (question 3): bing search results truncation.
>
> We use the snippets returned from bing which are relatively short so we do not perform additional truncation.

---

### Official Review · Reviewer_PjRU · 2023-08-05

**Typos Grammar Style And Presentation Improvements:** 1. Figure 2 caption has grammar errors
**Soundness:** 4

**Excitement:**

4: Strong: This paper deepens the understanding of some phenomenon or lowers the barriers to an existing research direction.

**Paper Topic And Main Contributions:**

This paper proposed a novel method for long-form retrieval-augmented generation. It is mainly motivated by two weaknesses in previous retrieval-augmented methods:
1. The past generated tokens do not necessarily reflect what information is beneficial for generating the next token. Therefore, this work proposed to first generate a "pseudo" next sentence and used this sentence as the retrieval query.
2. Using the past context within a sliding window does not always yields good queries for retrieval. This work proposed an active retrieval method which focuses on the low-confidence tokens in the generated sentence.
Besides, the authors also explored two methods of formulating the retrieval query using the next generated sentence with low-confidence tokens: (1) masking out low-confidence tokens with a probablity, or (2) re-generate a question with ChatGPT targeting on low-confidence spans.

**Questions For The Authors:**

A. Some descriptions in section 3.2 are not clear. In line 320, what does "extract all spans from s_t with probabilities below beta" mean? How do you define a "span" here? What is "the probability of a span"? What role is the hyper-parameter beta here?

B. In section 5, you mentioned the experiments are conducted in the manner of few-shot in-context learning. Do the few-shot examples have retrieved documents with them? Or do they only have questions and answers?

C. I didn't fully understand the setting in section 6.2 about the importance of forward-looking retrieval. When you retrieve with the next sentence in this experiment, did you "first generate the next sentence, then use the complete next sentence as the query, then re-generate the next sentence"?

D. In table 4, why don't you test beta=1? From my understanding, even when beta=1, it just means masking all low-confidence tokens instead of the whole sentence, right?

**Reasons To Accept:**

1. This paper is based on a novel idea: active retrieval-augmented generation. First, it leverages future information (the pseudo next sentence) to formulate the query instead of the past information. Second, it actively selects when to retrieve by targeting on the low-confidence tokens, which is more efficient than invoking retrieval in every sentence. These two perspectives as long as the proposed simple-and-effective solutions can bring insight to the field of long-form retrieval-augmented generation.
2. The proposed method is proved effective in multiple different tasks. The authors did comprehensive ablation studies to prove that both "using future information" and "using low-confidence tokens" are effective in the retrieval approach.

**Reasons To Reject:**

1. I didn't understand why section 3.1 (FLARE-instruct) is a type of "forward-looking retrieval augmentation". The description in section 3.1 and the prompt in E.1 does not convince me that this method leverages any anticipated information in the future. Given the poor performance of this method in table 1, I think this method should be listed as a baseline instead of part of the proposal.
2. The authors are claiming that "using information in the past" is not useful for predicting future tokens throughout the whole paper. in addition to indirect evidence from downstream accuracy scores, is there any direct experimental evidence on this (e.g., analysis on the relationship between the retrieved documents and the next generated sentence, or to what extent the model uses the retrieved information)? Besides, it would be better to have a baseline to use all the past information (including all generated tokens and the original query itself) as retrieval query.
3. Why aren't all three multi-time retrieval baselines in table 1 tested on all datasets? Especially for question decomposition, which is the strongest baseline in table 1.
4. The involvement of the external model ChatGPT in query generation may bring the risk of using additional knowledge of ChatGPT to assist the pipeline. Therefore, a safer option is to use the same model (text-davinci-003) for query generation.

**Reproducibility:**

4: Could mostly reproduce the results, but there may be some variation because of sample variance or minor variations in their interpretation of the protocol or method.

**Reviewer Confidence:**

4: Quite sure. I tried to check the important points carefully. It's unlikely, though conceivable, that I missed something that should affect my ratings.

---

> ### Author Rebuttal · Authors · 2023-08-28
>
> Thanks for the constructive feedback, and we are happy that you appreciated the idea!
>
> > Q1 (reason to reject 1): why FLARE-instruct is forward-looking retrieval augmentation?
>
> We use task-agnostic instructions and 3 exemplars (in Prompt E.1) to prompt the LM to generate search terms that reflect *future* information needs, thus can be forward-looking. For example, in the second exemplar, the search term “Ghana flag red meaning” is generated *before* explaining the meaning.
>
> > Q2 (reason to reject 1): FLARE-instruct as a baseline instead of part of the proposal.
>
> We indeed view FLARE-direct as our major technical contribution and view FLARE-instruct as an exploratory variation/ablation of active retrieval augmented generation. We will be sure to reflect this in the revised paper. Part of the reason why FLARE-instruct performs poorly is because it is hard to teach text-davinci-003 to generate proper search terms with just 3 exemplars. A promising future direction is to directly finetune LMs (such as LLaMA) to generate search terms while answering questions with high-quality training data.
>
> > Q3 (reason to reject 2): direct evidence on using information in the past is less useful.
>
> Thanks for the suggestion! We will add an analysis measuring the similarity (e.g., ROUGE) between documents retrieved based on the past/next sentence and the gold next sentence.
> We hypothesize that retrieval with the next sentence can find information more similar to the gold next sentence, therefore increasing the downstream performance, as reflected in table 3.
>
> > Q4 (reason to reject 2): a baseline that uses all the past information.
>
> In our preliminary experiments, we tried using different numbers of past tokens (as much as 128) as retrieval queries, and we found using 16 tokens (used in the paper) generally achieved the best overall accuracy. We hypothesize that tokens far in the past might not be relevant to the current context, and using all past tokens as queries might distract the model. We will add this ablation experiment in the revised version.
>
> > Q5 (reason to reject 3): why aren't the two baselines (previous-sentence and question decomposition) tested on the other 3 datasets?
>
> We do have the performance of the previous-sentence method on the other 3 dataset and it is slightly worse than the previous-window method, similar to the observation on 2WikiMultihopQA. For example, EM of the previous-sentence approach on StrategyQA/ASQA are 71.0/49.2. We will add these numbers in the revised version.
>
> The question decomposition method proposed in self-ask (Press et al., 2022), as the name suggests, specializes in and is proposed for solving compositional questions. They are questions that require multiple hops of querying and can straightforwardly break down into simpler questions. Therefore we reuse their manually designed prompts on multi-hop QA (i.e., 2WikiMultiHopQA) as a critical baseline. This helps us demonstrate FLARE works well compared to specialized prompting strategies. On the other hand, when it comes to the other datasets such as summarization, the way to decompose becomes unclear, and to the best of our knowledge, there is no such existing work. This is also why FLARE is a more versatile method that does not require special designs.
>
> > Q6 (reason to reject 4): use the same model (text-davinci-003) for query generation.
>
> Thanks for the suggestion! The reasons why we use text-davinci-003 for answer generation and chatgpt (gpt-3.5-turbo) for query generation are:
> 1. Answer generation requires LMs to return probabilities while query generation does not. Chatgpt does not provide probabilities, so we used text-davinci-003 for answer generation.
> 2. Chatgpt is much cheaper than text-davinci-003.
> 3. Chatgpt is better at following instructions to generate reasonable queries.
> We will consider adding a comparison experiment using text-davinci-003 to generate queries.
>
> > Q7 (question 1): what are spans used in query generation?
>
> We first find all low-probability tokens (lower than beta) and consecutive low-probability tokens form spans. Queries are generated for each span separately. beta decides how much we trust the LM’s initial generation. Lower beta (e.g., 0.1) means we are lenient and only mask or generate queries for tokens with probabilities < 0.1.
>
> > Q8 (question 2): do the few-shot examples have retrieved documents?
>
> For 2WikiMultihopQA, we follow previous baselines (Trivedi et al., 2022) to use retrieved documents for exemplars. For other datasets, due to the context length limit, we do not use retrieved documents for exemplars. This is mentioned in the appendix (line 1090-1095), and we will move it to the main body to make it clear.
>
> > Q9 (question 3): importance of forward-looking retrieval.
>
> Your understanding is correct, we first generate the next sentence, then use the complete next sentence as the query, then regenerate the next sentence.
>
> > Q10 (question 4): beta = 1 in table 4.
>
> Setting beta = 1 will mask all tokens in the sentence and nothing is left to be used as the retrieval query.
>
> > Q11: Typos in figure 2 and figure 4.
>
> Thanks for pointing out these presentation issues! We will correct typos and reorganize figure 4.

---

### Meta-Review · Area_Chair_ZnyS · 2023-09-18

**Recommendation:** 5

**Metareview:**

This paper proposes a new method, Active Retrieval, which prompts an LLM to trigger iterative retrieval while generations. Their experimental results show effectiveness on Multi-hop QA, long-form QA, and aspect-based summarization datasets. While I agree with some of the limitations shared by the reviewers (e.g., limited novelty given Toolformer, applicability to other LLMs, evaluations on focused tasks), overall the idea is exciting and the experiments are well conducted. Therefore, I recommend for acceptance.

---

### Decision · Program_Chairs · 2023-10-07

**Decision:**

Accept-Main

**Comment:**

This paper proposes a new method, Active Retrieval, which prompts an LLM to trigger iterative retrieval while generations. Their experimental results show effectiveness on Multi-hop QA, long-form QA, and aspect-based summarization datasets. While I agree with some of the limitations shared by the reviewers (e.g., limited novelty given Toolformer, applicability to other LLMs, evaluations on focused tasks), overall the idea is exciting and the experiments are well conducted. Therefore, I recommend for acceptance.